**∂ | Open Peer Review** | Bacteriology | Research Article

# Rescue of pyrimidine-defective *Pseudomonas aeruginosa* through metabolic complementation

Hafij Al Mahmud,[1] Randy Garcia,[1] Alexsis Garcia,[1] Jiwasmika Baishya,[1,2] Catherine A. Wakeman[1]

**ABSTRACT** Chronic infections harbor multiple pathogens where dynamic interactions between members of the polymicrobial community play a major role in determining the infection outcome. For example, in a nutrient-rich polymicrobial infection, bacteria have the potential to undergo evolutionary changes that impair their ability to synthesize essential metabolites. This adaptation may facilitate metabolic interdependencies between neighboring pathogens and lead to difficult-to-treat chronic infections. Our research group previously demonstrated that *Pseudomonas aeruginosa* (PA) and *Staphylococcus aureus* (SA), typically considered classical competitors, can adopt a cooperative lifestyle through bi-directional purine exchange medicated by exogenous DNA (eDNA) release. To further validate our initial findings, in this study, we investigated the potential exchange of pyrimidine between PA and other pathogens, which is another constituent of DNA. In our findings, we observed that a pyrimidine-deficient transposon mutant strain of PA showed improved growth when co-cultured with wild-type PA, SA, *Acinetobacter baumannii* (AB), and *Enterococcus faecalis* (EF). Additionally, improved fitness of pyrimidine-deficient PA was further observed in chemical complementation with eDNA and uridine-5′-monophosphate. Interestingly, the rescue of PA growth through eDNA complementation is not as effective as in intact cells, such as SA, AB, EF, and wild-type PA, implying that eDNA is a lesser contributor to this metabolic complementation. Also, the exchange mechanism between pathogens involves more active mechanisms beyond simple eDNA or metabolite release. Our data further highlights the importance of cell-to-cell contact for effective and increased metabolic complementation.

**IMPORTANCE** This research holds crucial implications for combating chronic infections, where multiple pathogens coexist and interact within the same environment. By uncovering the dynamic exchange of pyrimidines between *Pseudomonas aeruginosa* (PA) and *Staphylococcus aureus* (SA), our study reveals a previously unrecognized aspect of interspecies cooperation. The observed enhanced growth of a pyrimidine-deficient PA strain when co-cultured with SA suggests potential avenues for understanding and disrupting bacterial metabolic interdependencies in chronic infection settings. Furthermore, our findings highlight the mechanisms involved in metabolic exchange, emphasizing the importance of cell-to-cell contact. This research explored essential metabolic interactions to address the challenges posed by difficult-to-treat chronic infections.

**KEYWORDS** polymicrobial interactions, pyrimidine exchange, *Pseudomonas aeruginosa*, *Staphylococcus aureus*

Address correspondence to Hafij Al Mahmud, hafij-al.mahmud@ttu.edu, hafij.a.mahmud@gmail.com, or Catherine A. Wakeman, catherine.wakeman@ttu.edu.

Hafij Al Mahmud and Randy Garcia contributed equally to this article. The order of authorship was determined alphabetically.

The authors declare no conflict of interest.

See the funding table on p. 13.

Cystic fibrosis (CF) is a genetic disorder caused by a mutation in the CF trans-conductance regulator gene (1). The mortality and morbidity in CF patients are often associated with infection (2). CF-infected lung often harbors multiple pathogens,

including *Pseudomonas aeruginosa* (PA), *Staphylococcus aureus* (SA), *Hemophilus influenzae*, *Burkholderia cepacia* complex, and so on. Notably, the active multimodal interactions between these infecting pathogens and the surrounding environmental components, including the biofilm, can significantly alter the outcome of CF infection, similar to most chronic infections (3). These interactions can be competitive, where one microorganism can outcompete other neighboring organisms by producing lethal molecules or hijacking essential nutrients. For example, iron depletion in co-culture may increase the lysis of SA by PA through secreted 2-alkyl-4(1H)-quinolones (4). Bacterial co-culture on human bronchial epithelial cell monolayers showed PA drives SA metabolism from aerobic respiration to fermentation and eventually kills SA by secreting siderophores or 2-heptyl-4-hydroxyquinoline N-oxide (HQNO) (5). The interactions between pathogens can also be cooperative, where pathogens can help in the survival of the other co-infecting pathogens through complementing nutritional deficiencies and protecting against host immune proteins or antibiotic stresses (6). For example, PA isolated from coinfected patients is found to be less competitive against SA; in fact, PA with mucoid phenotype would become severely inactive against SA and would reside within the infection site together. Alginate-producing mucoid strains of PA downregulate the synthesis of different virulence factors essential for the killing of SA (7). In the polymicrobial lung infection model, virulence factors secreted by SA are found to be helpful in the proliferation, spread, and pathogenicity of gram-negative pathogens like PA through compromising host immunity (8). In addition, host immune proteins like calprotectin may facilitate the co-colonization of these two classical competitors, PA and SA, in cystic fibrosis lung (9).

In this study, we are particularly interested in the cooperation between CF infection pathogens PA and SA in terms of nutritional complementation. In chronic infections, co-infecting pathogens or commensal bacteria might lose the ability to synthesize essential biomolecules because of random mutation and become dependent on the neighboring organisms or host (10). The evolution of auxotrophic PA is common in CF infection. For example, methionine-dependent PA auxotrophy can evolve due to the presence of methionine in the environment as di-, tri-, tetra-, and pentapeptides (11, 12). Furthermore, our previous research has documented the metabolic complementation among CF pathogens through purine cross-feeding. Specifically, our findings provided evidence for the reciprocal exchange of purine compounds, fostering cooperative interactions between well-known rivals, PA and SA (13).

Pathogenic bacteria rely on *de novo* nucleotide biosynthesis to initiate infection, survive, and be virulent. Regulators of this process have been shown to be essential in regulating the production of virulence factors (14, 15). Synthesis or acquisition of purines and pyrimidines is essential for cellular functions and the reproduction of microorganisms (14). For example, pyrimidine biosynthetic genes are essential for PA to grow well in the CF-infected lung environment (16). In *de novo* purine biosynthesis, precursor molecule 5-phosphoribosyl-α-1-pyrophosphate (PRPP) is converted into the final product inosine-5′-monophosphate (IMP) by the action of different enzymes, encoded by genes such as *purF*, *purD*, *purN*, *purT*, *purS*, *purQ*, *purI*, *purM*, *purK*, *purE*, *purC*, *purB*, *purH*, and so on (14). Similarly, two methods are employed by organisms for obtaining pyrimidines: *de novo* synthesis, which is a universal process consisting of six consecutive enzyme reactions, or a salvage pathway (17). In *de novo* pyrimidine biosynthesis, the precursor molecule carbamoyl phosphate (CP) is converted into the final product uridine-5′-monophosphate (UMP) by the action of different enzymes, namely carbamyl phosphate synthetase, aspartate transcarbamylase, dihydroorotase, dihydroorotate dehydrogenase, orotate phosphoribosyltransferase, and orotidylate decarboxylase (18, 19). These enzymes are encoded by six unlinked genes, namely *pyrB*, *purC*, *pyrK*, *pyrD*, *pyrE*, *pyrF*, and so on (14). Finally, the molecule UMP can be converted to different pyrimidines, such as UDP, UTP, CTP, and so on, by the action of other enzymes. Apart from these, UMP can be produced from pyrimidine bases and nucleosides by enzymatic action through the salvage pathway (17). Among those essential genes required for *de*

*novo* pyrimidine biosynthesis, *pyrB*, *pyrD*, *pyrE*, and *pyrF* genes have been selected for this study, as the products of these genes play a significant role in pyrimidine biogenesis (15). This research study examines how the deficient organism can obtain pyrimidine through the utilization of secreted eDNA or secreted nucleotides from neighboring pathogens, thereby complementing its pyrimidine needs. As well as, we studied the role of cell-to-cell contact in efficient pyrimidine exchange between pathogens.

## MATERIALS AND METHODS

### Bacterial strains and culture

Laboratory reference strains of *P. aeruginosa* UCBPP-PA14 (PA14), *S. aureus* USA300 (JE2), *Enterococcus faecalis* ATCC 29212 (EF), and *Acinetobacter baumannii* ATCC 19606 (AB) were used as the control wild-type strains in this study. To study pyrimidine complementation in PA, four pyrimidine-defective PA strains (*pyrB::tn*, *pyrD::tn*, *pyrE::tn*, and *pyrF::tn*) were used. These transposon mutant strains of PA were taken from a non-redundant library. A marinar-based transposon marxt7 was used to create the non-redundant library of PA14 transposon mutants (20). One of the mutants, *pyrB::tn*, was confirmed via whole genome sequencing performed by Plasmidsaurus. We acquired the FASTA sequence data of the *pyrB::tn* strain from Plasmidsaurus. This sequence was then aligned with the genome of the wild-type strain PA14 using Mauve, confirming the presence of a transposon insert in the *pyrB* gene of the mutant strain. Both the wild-type PA, EF, AB, and transposon mutant strains of PA, were cultured in Lysogeny broth (LB), and the JE2 strain was cultured in Tryptic soy broth (TSB).

### Pyrimidine cross-feeding between bacteria in liquid culture

Bacterial cells were grown in the pyrimidine-deficient minimal growth medium Roswell Park Memorial Institute (RPMI) supplemented with 1% casamino acid (CA) as a carbon source to study pyrimidine cross-feeding. Overnight cultures of PA, AB, EF, and JE2 at 37°C in LB or TSB were washed in sterile 1× phosphate-buffered saline (PBS). Following the PBS wash, respective cells were normalized to a cell density of ~$10^8$ CFUs/mL (OD$_{600}$ = 1.0). Following normalization, the cells were inoculated in RPMI + 1% CA as mono and/or mixed cultures in 96-well plates to achieve a final density of ~$10^6$ CFUs/mL. The plates containing mono and mixed cultures were incubated at 37°C statically for 48 h. Following incubation, cells were serially diluted and spotted on selective media. Pseudomonas isolation agar and Mannitol salt agar plates were used as selective media for isolating and estimating the viable differential number of PA and JE2, respectively. Meanwhile, leads agar and enterococcosel agar was used as selective media for isolating and counting the number of viable AB and EF, respectively.

### Quantifying the abundance of eDNA in mono and mixed bacterial cultures

To estimate the concentration of secreted eDNA in mono and mixed bacterial cultures, overnight-grown bacterial cells were washed three times in 1× PBS. The cells were normalized to OD$_{600}$ of 1.0, which gives us a cell density of ~$10^8$ CFUs/mL. Cells were then inoculated as mono and/or mixed culture in RPMI + 1% CA to achieve a final density of ~$10^6$ CFUs/mL in a 96-well plate. Plates were incubated and grown at 37°C for 48 h statically. Following incubation, the concentration of eDNA in mono and mixed cultures was estimated using a dsDNA binding reagent Quant-iT PicoGreen (Invitrogen, VA, USA) by following the manufacturer's instructions with slight modifications. Generally, picogreen is incapable of penetrating bacterial cell membranes; therefore, we expect that it would only bind with the eDNA in the bacterial culture (21). Briefly, equal volumes of bacterial cultures were mixed with 200 times diluted picogreen reagent in 96-well plates. These plates were then incubated for 5 min at room temperature, following which the fluorescent intensity was detected using a multiplate biotek reader (Biotek Synergy H1), *Ex*/*Em* 485/528. The fluorescent intensity readings were converted to eDNA

concentrations using a standard curve. The standard curve was generated by measuring the fluorescent intensities of known concentrations (0–900 µg/mL) of eDNA in respective bacterial monocultures in RPMI media.

## Pyrimidine complementation in PA via eDNA

To validate the role of pyrimidine in rescuing the growth of pyrimidine-defective mutants, the cells were chemically complemented with herring sperm DNA as a source of exogenous pyrimidine. Briefly, wild-type and mutant cells were grown overnight, followed by three times wash with sterile 1× PBS. Then, the cells were normalized to a cell density of $10^8$ CFUs/mL. Normalized cells were diluted 100 times in RPMI + 1% CA media in 96-well plates preoccupied with various concentrations of eDNA. To observe complementation, two sets of experiments were done. In the first set, the eDNA was subjected to enzymatic digestion by DNAse at 37°C for an hour. In the second set, the eDNA was left undigested, and the following concentrations of eDNA, 0, 10, 100, 300, 600, and 900 µg/mL were tested for both the digested and undigested conditions. Cells with or without eDNA in 96-well plates were incubated for 48 h at 37°C statically. Following incubation, the cells were diluted serially and spotted on selective media to estimate the viable bacterial number.

## Pyrimidine complementation in PA via UMP

To further support the role of pyrimidine in rescuing the growth of pyrimidine-defective mutants, cells were chemically complemented with UMP, the product in *de novo* pyrimidine biosynthesis as a source of exogenous pyrimidine. Briefly, wild-type and mutant PA cells were grown overnight; then, the cells were washed three times with sterile 1× PBS. Next, the cells were normalized to a cell density of $10^8$ CFUs/mL with an optical density of 1 at 600 nm. Normalized cells were diluted 100 times in RPMI + 1% CA media in 96-well plates preoccupied with various concentrations of UMP. Cells with or without UMP (0–10 mg/mL) in 96-well plates were incubated for 48 h at 37°C statically. Following incubation, the cells were diluted serially and spotted on selective media to estimate the viable bacterial number.

## Rescue in pyrimidine-defective mutants of PA via JE2 cell-free supernatant

We cultured both wild-type and defective mutants in the presence or absence of JE2 cell-free supernatant. Briefly, JE2 cells were cultured overnight in RPMI + 1% CA at 37°C, and the culture was then filtered through a 0.22-µM filter to get cell-free supernatant. Wild-type and mutant PA cells were grown overnight; then, the cells were washed three times with sterile 1× PBS. Next, the cells were normalized to a cell density of $10^8$ CFUs/mL with an optical density of 1 at 600 nm. Normalized cells were diluted 100 times in RPMI + 1% CA media in 96-well plates preoccupied with 50% or 0% of JE2 cell-free supernatant. Cells were incubated for 48 h at 37°C statically. Following incubation, the cells were diluted serially and spotted on selective media to estimate the viable bacterial number.

## Pyrimidine cross-feeding in transwell plates

Bacterial cells were grown in the pyrimidine-deficient minimal growth medium RPMI supplemented with 1% CA as a carbon source. Overnight cultures of PA14, *pyrB::tn*, and JE2 at 37°C in LB were washed two times in sterile 1× PBS. Following the PBS wash, respective cells were normalized to a cell density of ~$10^8$ CFUs/mL (OD$_{600}$ = 1.0). Following normalization, the cells were inoculated in RPMI + 1% CA as mono and/or mixed cultures in transwell (Corning Incorporated, USA) plates to achieve a final density of ~$10^6$ CFUs/mL. Briefly, transwell contained 100 µL of media with or without bacterial cells, whereas the plate well contained 600 µL of media with or without bacterial cells. In JE2-*pyrB::tn* co-culture, JE2 cells were cultured in the transwell, whereas PA cells were cultured in the plate well. In the case of PA14-*pyrB::tn* co-culture, PA14 was cultured in the transwell. The plates containing mono and mixed cultures were incubated at 37°C

statically for 48 h. Following incubation, the cells were serially diluted and spotted on selective media. Pseudomonas isolation agar and Mannitol salt agar plates were used as selective media for isolating and estimating the viable differential number of PA and JE2, respectively.

## RESULTS AND DISCUSSION

### Rescue of pyrimidine-deficient PA by other pathogens through metabolic complementation

PA and SA are two major pathogens frequently isolated from chronic infection niches such as CF lungs, chronic wounds, foot ulcers, and so on (22–24). In normal laboratory conditions, PA usually outcompetes SA in mixed cultures, whereas these classical competitors are often isolated together from different chronic infections (25). Chronic infections, including CF infection, often exhibit a very complex microenvironment that can alter bacterial physiology and ultimately alter the disease outcome. For example, pathogenic microorganisms may evolve to become auxotrophic by the influence of diverse factors in a chronic environment (26). Auxotrophic PA is isolated from CF infections, and that may be defective in synthesizing different essential metabolites (27). These metabolic deficiencies can be complemented through sharing resources by neighboring pathogens. In one of our studies, we reported how purine, an essential metabolite, can be cross-fed between PA and SA through the secretion of exogenous DNA by metabolically active cells (13). We know that DNA comprises purines and pyrimidines, two of the crucial metabolites required for the growth and colonization of bacteria. Therefore, we hypothesize that neighboring pathogens can complement pyrimidine molecules in auxotrophic bacteria in a manner similar to purine complementation. To support this hypothesis, we have selected several pyrimidine-deficient (*pyrB::tn*, *pyrD::tn*, *pyrE::tn*, and *pyrF::tn*) transposon mutant strains. Each mutant strain is defective for a unique enzyme (aspartate carbamoyl transferase, dihydroorotate dehydrogenase, orotate phosphoribosyl transferase, and orotidine-5′-monophosphate decarboxylase enzyme names, respectively) that is essential for the *de novo* biosynthesis of pyrimidine in bacteria. To study pyrimidine complementation, we co-cultured these mutant strains together with a wild-type lab strain PA14 as a control with a wild-type lab strain of SA, JE2, in RPMI + 1% CA minimal media. We used RPMI + 1% CA due to the absence of nutrients or metabolites other than CA in it, which is essential to investigate pyrimidine complementation in mixed cultures. Following incubation, we found that all the auxotrophic pyrimidine-deficient PA cells are defective in growth in monoculture compared to wild-type PA14. Whereas in mixed cultures, the presence of JE2 significantly enhanced the fitness of PA mutants compared to their wild-type counterparts ($P < 0.0001$), resulting in a substantial increase in their population. In fact, the mutant cells proliferate to the same extent as the wild-type cells (Fig. 1A).

Considering the differential growth of mutant cells compared to PA14 in monoculture, we normalized their fitness in mixed cultures with their respective growth in monocultures. These findings indicate a significant rescue of all mutant cells (*pyrB::tn*; $P < 0.0005$, *pyrD::tn*; $P < 0.005$, *pyrE::tn*; $P < 0.005$, *pyrF::tn*; $P < 0.05$) by JE2 in mixed cultures, surpassing the rescue observed in PA14 while co-cultured with JE2 (Fig. 1B).

These data demonstrate that JE2 can rescue the pyrimidine-deficient PA in mixed cultures. This rescue might be achieved through the release of eDNA, nucleosides, or nitrogenous bases. The release of eDNA or other secretory molecules in the liquid cultures may be a result of cellular degradation following lysis and/or secretion by metabolically active bacterial cells. Furthermore, we estimated the fitness of JE2 in mono and mixed cultures (Fig. 1C). The number of JE2 cells in mixed culture with PA14 reduced significantly ($P < 0.0005$) compared to the number of JE2 in monoculture. This is reasonably expected because PA14 is known to outcompete JE2 *in vitro* under laboratory conditions. However, the fitness of JE2 in mixed cultures with all the pyrimidine-defective mutants did not reduce significantly. In fact, it shows a similar fitness of JE2 in

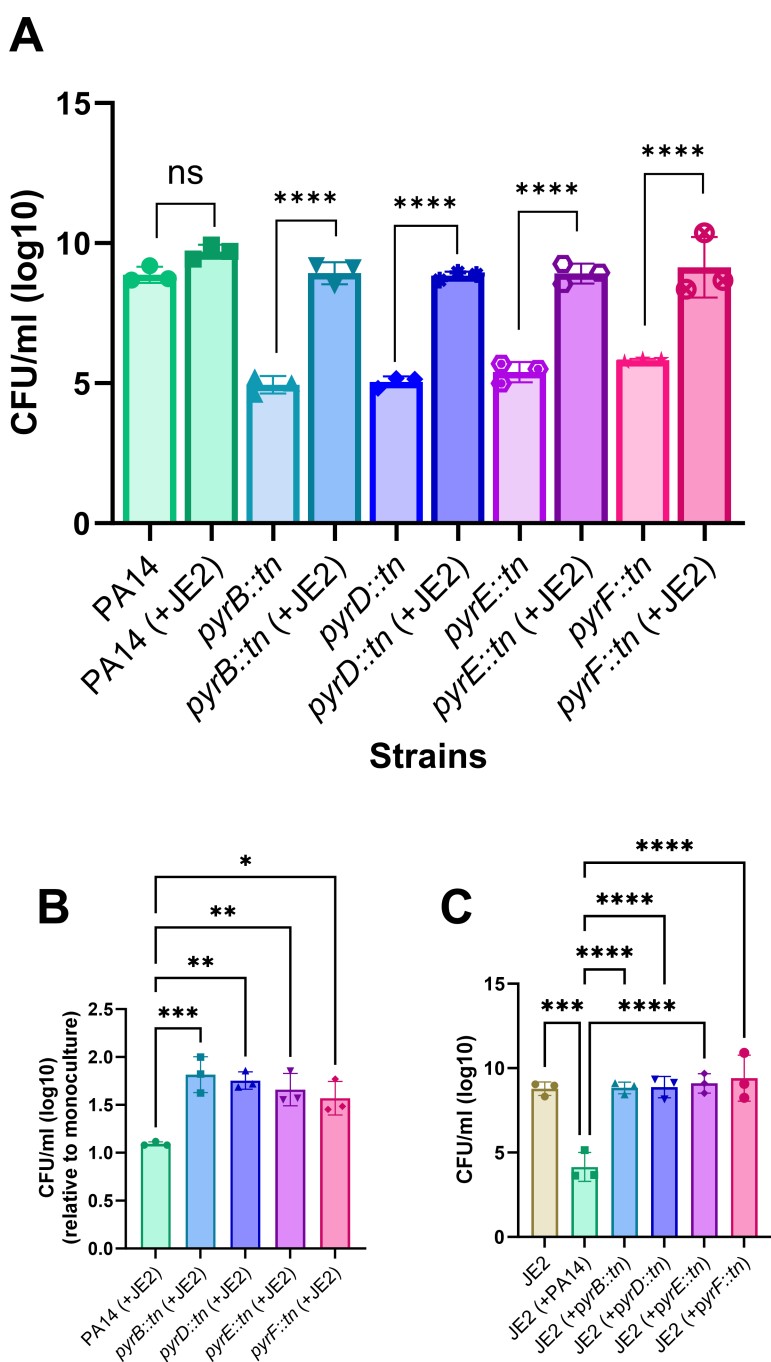

**FIG 1** Rescue of pyrimidine-deficient mutants of *P. aeruginosa* by *S. aureus*. Fitness of *P. aeruginosa* (both wild-type and pyrimidine-defective mutants) in monoculture and mixed culture with *S. aureus* (A), the fitness of *P. aeruginosa* in mixed culture with *S. aureus* compared to their respective monocultures (B), the fitness of *S. aureus* in monoculture and in mixed cultures with *P. aeruginosa* (C). Error bar represents the SD of data obtained from three biological replicates conducted on different days. Each biological replicate consisted of three technical triplicates per day. "*" designates $P < 0.05$, "**" designates $P < 0.005$, "***" designates $P < 0.0005$, "****" designates $P < 0.0001$ as depicted by one-way ANOVA, and ns denotes not significant ($P > 0.05$).

monoculture and a significant increase ($P < 0.0001$) compared to JE2 with PA14. This increased fitness of JE2 in mixed cultures with pyrimidine-deficient mutants may be attributed to the fact that the auxotrophic strains are less competitive against JE2

 

compared to PA14 (15). Overall, the metabolites mediating interspecies complementation may, at least in part, be released from lysed cell populations.

Further, we wanted to explore whether the observed metabolic complementation of pyrimidine-defective PA is restricted to SA only or not. To this extent, we conducted similar experiments using AB, a representative gram-negative pathogen, and EF, a representative Gram-positive pathogen. Both AB and EF are associated with different life-threatening infections (28, 29). Pyrimidine-defective mutant *pyrB::tn* was selected as a representative auxotrophic mutant for this experiment. Furthermore, we confirmed the *pyrB::tn* mutant to be truly deficient in pyrimidine biosynthesis using chemical complementation with uridine-5′-monophosphate (Fig. S1). Like earlier mixed culture experiments with JE2, the fitness or PA14 was found to be unaffected in mixed culture with AB and EF compared to monoculture. Whereas, compared to monoculture, a significant ($P < 0.0001$) rescue in *pyrB::tn* cells has been found in the presence of both AB and FE (Fig. 2A). That tells us that the growth rescue of the pyrimidine-defective mutants is not restricted to SA or Gram-negative or Gram-positive pathogens. Neighboring pathogens with functional pyrimidine biosynthesis machinery may rescue pyrimidine-defective mutants of PA in infection sites. Furthermore, both the wildtype PA14 and *pyrB::tn* mutant were found to be slightly competitive ($P < 0.05$) against AB, whereas EF was found to be unresponsive to PA14 and *pyrB::tn* mediated killing in mixed culture (Fig. 2A). That tells that the molecules responsible for the pyrimidine complementation may come from metabolically active or lysed cells.

Similar to AB and EF, *pyrB::tn* growth was found to be significantly ($P < 0.05$) rescued by the PA14 wild-type in mixed culture separated by a membrane (Fig. 2B). On the other hand, the fitness of PA14 remains unaffected by the presence of *pyrB::tn*. In summary, these data emphasize that metabolites secreted by wild-type PA14 can complement pyrimidine deficiency in defective mutants. Additionally, intraspecies complementation can be achieved without cell-to-cell contact.

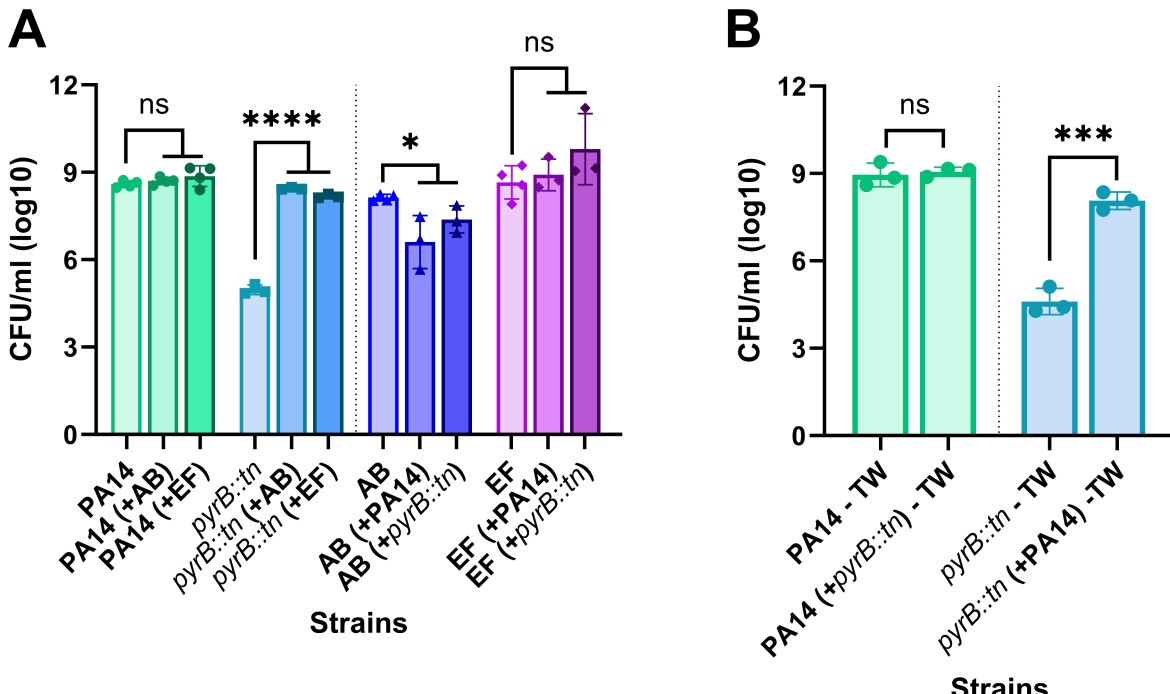

**FIG 2** Rescue of pyrimidine-deficient mutants of *P. aeruginosa* by *A. baumannii* and *E. faecalis*. (A) Fitness of *P. aeruginosa pyrB::tn* in monoculture and mixed culture with *A. baumannii* and *E. faecalis*. (B) Fitness of PA14 and *pyrB::tn* in monoculture and in mixed culture in *trans* well condition. The error bar represents the SD of data obtained from three biological replicates conducted on different days. Each biological replicate consisted of three technical triplicates per day. "AB" denotes *A. baumannii,* "EF" denotes *E. faecalis,* "*" designates $P < 0.05$, "****" designates $P < 0.0001$ as depicted by a two-tailed unpaired Student's *t* test, ns denotes not significant ($P > 0.05$), and TW denotes cells were cultured in transwell.

 

## Release of eDNA by CF pathogens

Our earlier experiments showed that neighboring pathogens like SA can rescue pyrimidine-deficient mutants of PA. Our hypothesis proposes that SA can supply pyrimidine to pyrimidine auxotrophic PA through the release of eDNA together with other secretory metabolites. To support this hypothesis, we have cultured wild-type PA14 and the pyrimidine-defective mutants of PA alone and together with wild-type SA, JE2, in RPMI + 1% CA media. Following incubation, the presence of eDNA was measured using the DNA staining dye picogreen. Our results depict that the presence of eDNA in JE2 monoculture is higher than in PA's pyrimidine-defective mutants (*pyrB::tn; P > 0.05, pyrD::tn; P < 0.05, pyrE::tn; P < 0.05, pyrF::tn; P > 0.05*) which might be attributed to the low growth of the defective mutants in monoculture (Fig. 3). Different microorganisms, including bacteria, may release eDNA in the environment through various mechanisms, such as by membrane vesicles or following cell lysis (30, 31). We had made similar observations in our previous study, where we showed the role of secreted eDNA from neighboring microbes in rescuing purine defective pathogenic microbes (13). Overall, the data from Fig. 3 suggest that eDNA is present in the culture supernatant and implies that microbe-derived eDNA also contributes to the overall eDNA present in the infection niches. In addition, the concentration of eDNA in wild-type culture supernatant is higher in comparison to defective mutants.

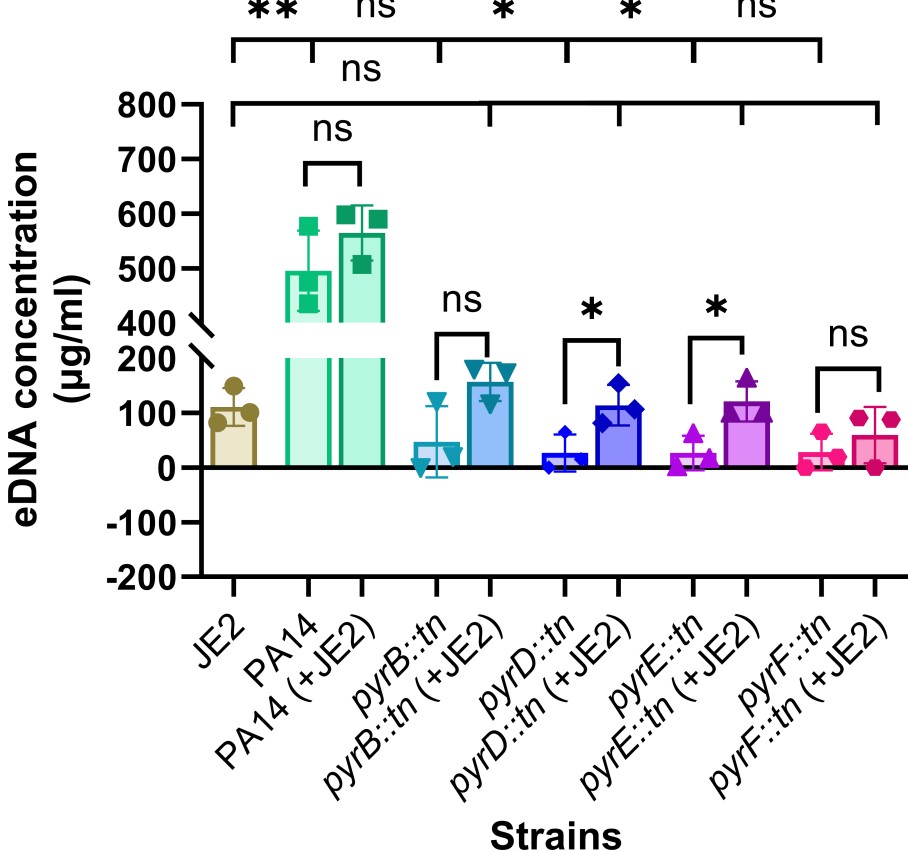

**FIG 3**  Abundance of eDNA in mono versus mixed cultures. Error bars represent the SD of data obtained from three biological replicates conducted on different days. Each biological replicate consisted of three technical triplicates per day. "*" designates *P* < 0.05 as depicted by a two-tailed unpaired Student's *t* test, and ns denotes not significant (*P* > 0.05).

## Pyrimidine complementation between PA and SA through bacterial secretory molecules such as eDNA

Through the study mentioned above, we have hypothesized that SA can rescue the growth of pyrimidine-defective PA by secreting eDNA. To support this hypothesis, we chemically complemented the pyrimidine-defective mutants of PA using eDNA to see their increased fitness benefit. Interestingly, CF-infected lungs are enriched in eDNA (up to 900 µg/mL), which may be derived from the infecting pathogens as well as the infected host tissues (32). In this experiment, we cultured both the wild-type PA14 and their pyrimidine-defective mutants with various concentrations of eDNA in RPMI + 1% CA media. The range of eDNA concentrations used for the complementation was selected based on the reported concentration of eDNA in chronically CF-infected lungs. Furthermore, we conducted this chemical complementation reaction to study the role of the eDNA molecules in the metabolic complementation reaction with both enzymatically digested and undigested eDNA. Following co-incubation with none to various concentrations of digested or undigested eDNA, the fitness of each auxotrophic PA was evaluated by comparing the colony-forming units per mL in different treatment conditions.

In the presence of both digested and undigested eDNA, all the pyrimidine-defective mutants showed a concentration-dependent increase in their fitness compared to no treatment control (Fig. 4A and C). Interestingly, the increased fitness in pyrimidine-defective mutant growth is statistically more significant when treated with enzymatically digested eDNA versus the undigested one. This can be due to the abundance of free pyrimidine molecules in an environment treated with endonuclease enzymes. Furthermore, the fitness of wild-type PA14 was found to be stable irrespective of eDNA treatment. PA14 is not defective for pyrimidine biosynthesis; therefore, eDNA does not influence their fitness as it does for their auxotrophic counterparts. Another interesting finding is that, unlike purine complementation, pyrimidine-defective mutants' growth did not reach the level of wild-type PA14 when treated with a maximum concentration of eDNA (900 µg/mL). The reason for this could be that pyrimidine mutants are less effective in absorbing pyrimidine compared to the absorption of purines by purine-defective mutants. Overall, this emphasizes that eDNA might be a minor contributor to pyrimidine complementation in PA.

As pyrimidine is an essential molecule for the growth and survival of bacteria, defective mutants are naturally growth-deficient compared to the wild-type PA14 strain. Therefore, to consider the reduced growth of pyrimidine-defective mutants, we normalized the growth of both the wild-type and the auxotrophic strains' fitness in the presence of eDNA with fitness in the no eDNA control. The relative fitness data show that the growth of pyrimidine-defective mutants has increased in the presence of both enzymatically digested and undigested eDNA (Fig. 4B and D). In contrast, the fitness of PA14 was found to be stable in the presence of eDNA in both conditions.

Later, to get an idea of the most efficient metabolite exchange mechanism, we compared the rescue of pyrimidine-defective PA by eDNA to that of JE2. Our data showed that the fitness of PA14 remained the same irrespective of the presence of JE2 whole cell or eDNA. Unlike PA14, the fitness of pyrimidine-defective cells increased by at least 100 folds when co-cultured with JE2 rather than eDNA (Fig. 5). Interestingly, the concentration of eDNA released in PA-JE2 co-culture is way low compared to the maximum concentration (900 µg/mL) of eDNA used in the chemical complementation. However, when it comes to rescuing pyrimidine-defective strains, whole cells exhibit a higher efficacy compared to the presence of eDNA. That indicates that the exchange of eDNA or other metabolites is likely mediated by other mechanisms than simple eDNA or other metabolites released in the environment.

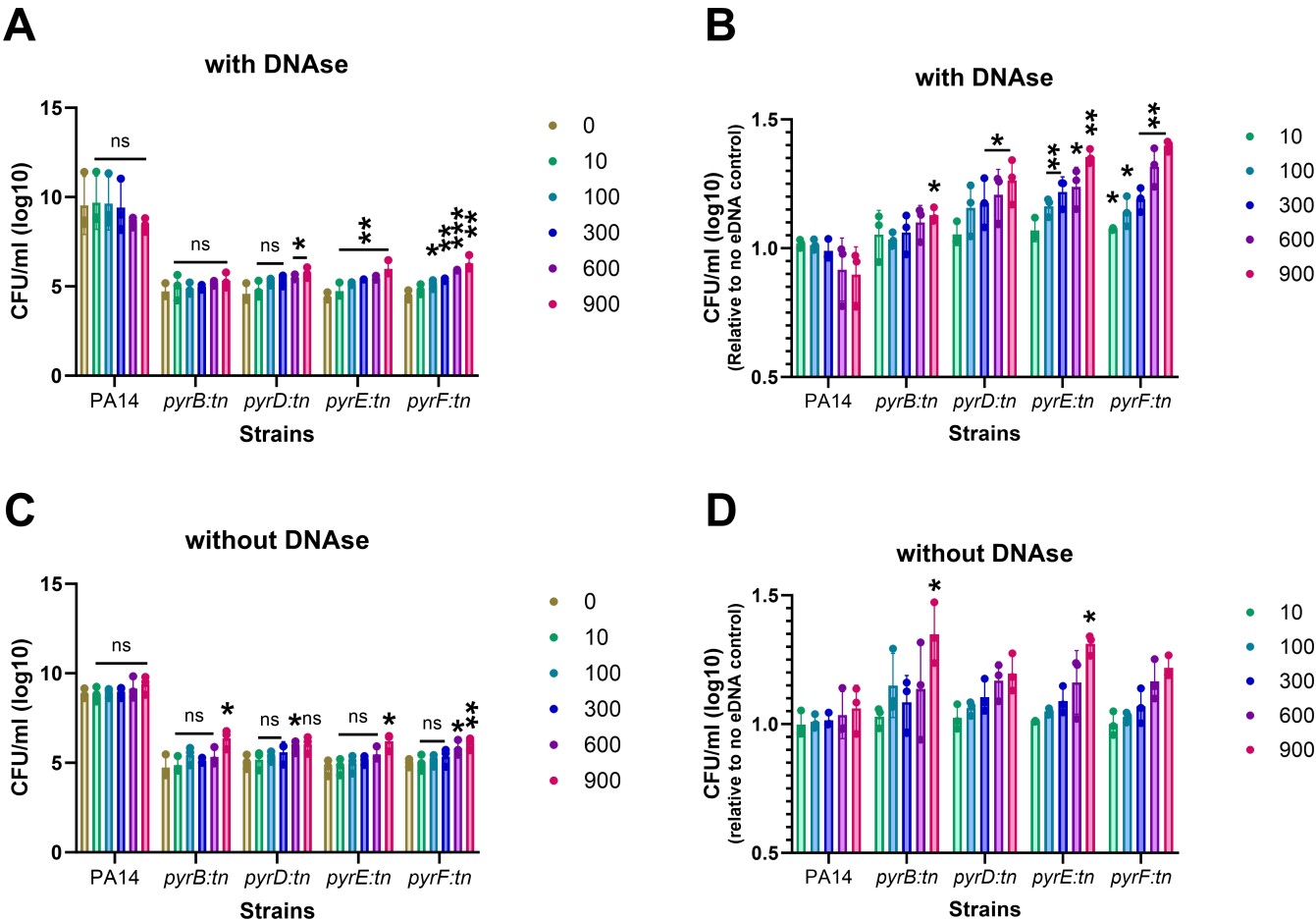

**FIG 4** Rescue of pyrimidine-deficient mutants of *P. aeruginosa* through eDNA. The lab reference strain, PA14, and the pyrimidine-deficient mutants of PA were exposed to various concentrations (0–900 µg/mL) of exogenous DNA following DNAse enzymatic digestion (A, B) or, without enzymatic digestion (C, D). The fitness of each PA strain was evaluated in the presence of eDNA (A, C). The fitness of each strain compared to no-eDNA control was evaluated in the presence of eDNA (C, D). Error bars represent the SD of data obtained from three biological replicates conducted on different days. Each biological replicate consisted of three technical triplicates per day. "*" designates $P < 0.05$ as depicted by a two-tailed unpaired Student's $t$ test, and ns denotes not significant ($P > 0.05$).

## Direct cell-to-cell contact facilitates the rescue of pyrimidine-deficient mutants of PA

Based on the findings observed in Fig. 4 and 5, we hypothesized that cell-to-cell contact is essential and could further facilitate the exchange of eDNA or other metabolites between the pathogens in the host-pathogen interface. To this end, we conducted this experiment to investigate whether cell-to-cell contact is essential for rescuing the pyrimidine-defective mutants of PA when co-cultured with JE2, either in direct contact or separated by a membrane. The results revealed a significant improvement in the fitness of pyrimidine-defective mutant *pyrB::tn* in both scenarios of co-culturing with the JE2 strains, compared to the fitness of PA14 in co-culture. Interestingly, this rescue was more pronounced when the two strains were in direct contact, as evidenced by the lower fitness of *pyrB:tn* when the strains were separated by a membrane (Fig. 6A). This implies that cell-to-cell contact is necessary for the complete rescue of pyrimidine-defective mutants of PA. On the other hand, both PA14 and *pyrB::tn* exhibited reduced competitiveness against JE2 when separated by a membrane compared to direct contact (Fig. 6B). That means direct contact may facilitate the cell lysis which may further contribute to the increased cell-to-cell contact mediated increased rescue in *pyrB::tn*.

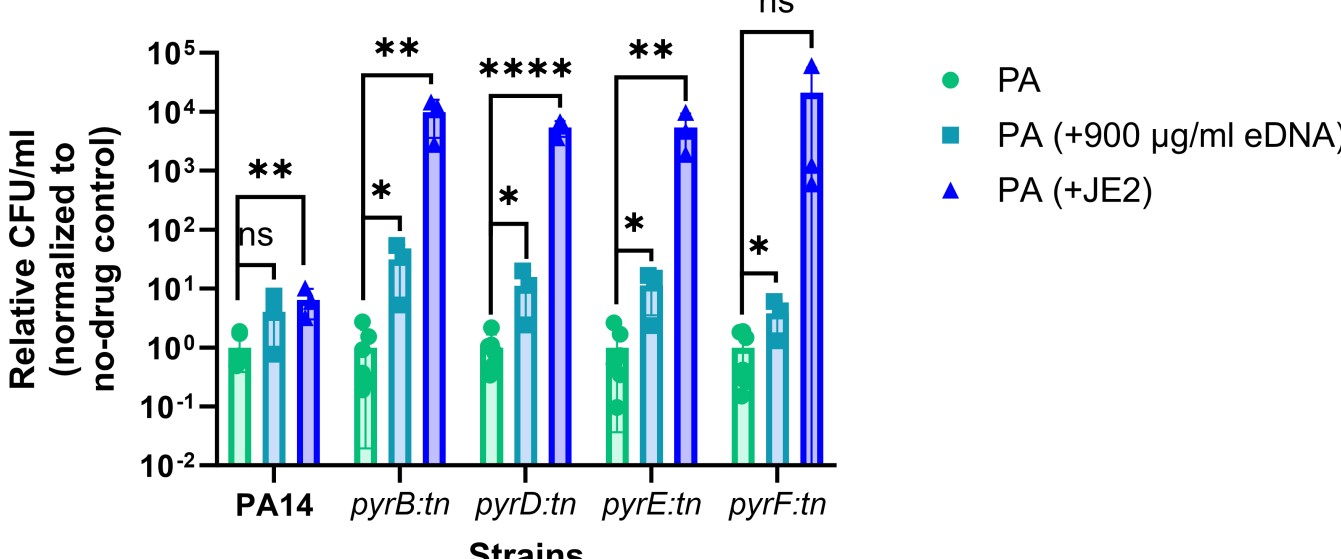

**FIG 5** Rescue of pyrimidine-deficient mutants of *P. aeruginosa* through co-culture versus chemical complementation. The lab reference strain, PA14, and the pyrimidine-deficient mutants of PA were exposed to JE2 and exogenous DNA (900 µg/mL). The fitness of each PA strain was evaluated in the presence of JE2 and eDNA. Error bars represent the SD of data obtained from three biological replicates conducted on different days. Each biological replicate consisted of three technical triplicates per day.

In this stage, we also wanted to see if the JE2 secretion is capable of rescuing the pyrimidine-defective mutants in the absence of metabolically active cells. To achieve this, we filter sterilized JE2 supernatants and then culture PA14 and *pyrB:tn* cells with or without the presence of JE2 supernatants. Our data showed that JE2 supernatant does not affect the fitness of PA14, but it significantly ($P < 0.05$) increases the fitness of *pyrB::tn* cells (Fig. 6C). Interestingly, similar to eDNA, the rescue of pyrimidine-defective mutants mediated by JE2 supernatant is only partial compared to the complete rescue observed in mixed culture with JE2. JE2 supernatant may contain eDNA, pyrimidine nucleotides, pyrimidine nucleosides, or nitrogenous bases that can complement the pyrimidine deficiencies in auxotrophic bacteria.

Overall, these data demonstrate that JE2 can complement pyrimidine deficiency in pyrimidine-defective PA, and cell-to-cell contact may further facilitate or expedite the complementation process.

## Conclusion

The results of this work suggest that auxotrophic bacterial growth and survival in chronic infection settings could be significantly influenced by interspecies metabolic complementation. Both metabolically active and lysed cells may facilitate the metabolic complementation in neighboring pathogens. Importantly, secreted eDNA is a minor contributor in metabolic complementation towards rescuing the pyrimidine-defective PA, but neighboring cells (SA) are more proficient for metabolic exchange via cell-to-cell contact mechanism. Apart from eDNA, secreted nucleotides, such as UMP, a precursor for pyrimidine nucleotide, may also rescue pyrimidine defective mutants. The role of secreted nucleosides and nitrogenous bases in this complementation requires further investigation. This study emphasizes how crucial it is to consider the intricate microenvironment of chronic infections when developing new therapies. Overall, this research suggests developing new strategies to combat chronic infections, such as targeting the pathways involved in the metabolic cross-feeding of pathogens.

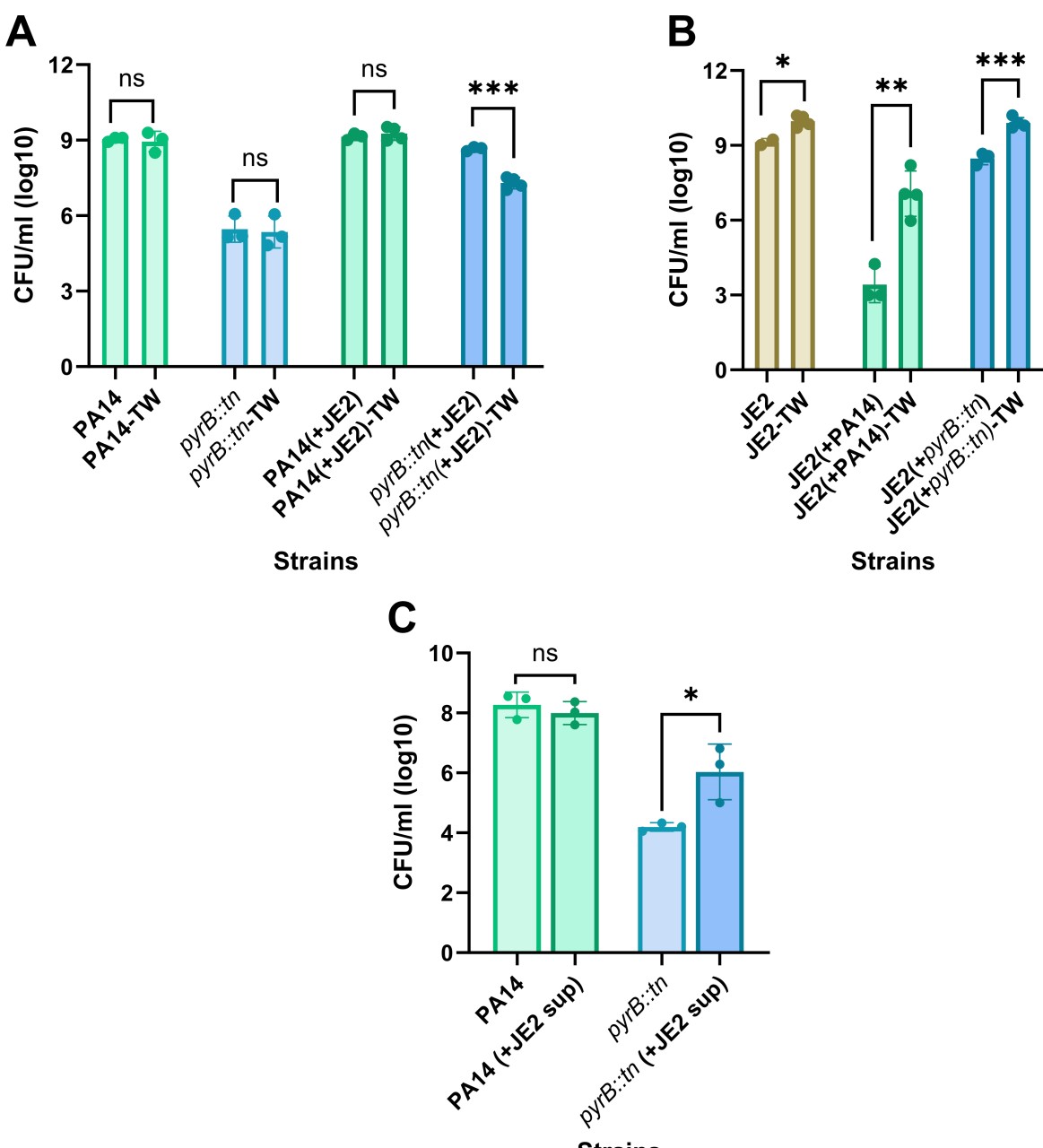

**FIG 6** Cell-to-cell contact facilitates the rescue of pyrimidine-deficient mutants of *P. aeruginosa* in mixed culture. The lab reference strain, PA14, and the pyrimidine-deficient mutants of *pyrB::tn* were cultured alone or together with JE2. In mixed culture, JE2 and PA cells were either cultured together or separated by a membrane. (A) The fitness of PA14 and *pyrB::tn* in monoculture and mixed culture. (B) The fitness of JE2 in monoculture and mixed culture. (C) Fitness of PA14 and *pyrB:tn* in the presence of JE2 cell-free supernatant. Error bars represent the SD of data obtained from three biological replicates conducted on different days. Each biological replicate consisted of three technical triplicates per day. "*" designates $P < 0.05$ as depicted by a two-tailed unpaired Student's *t* test, and ns denotes not significant ($P > 0.05$). TW; cells were cultured in transwell.

## AUTHOR AFFILIATIONS

[1]Department of Biological Sciences, Texas Tech University, Lubbock, Texas, USA
[2]The Assam Royal Global University, Guwahati, Assam, India

## AUTHOR ORCIDs

Hafij Al Mahmud  http://orcid.org/0000-0002-8944-493X

Catherine A. Wakeman 🔾 http://orcid.org/0000-0003-0311-6669

## FUNDING

| Funder | Grant(s) | Author(s) |
|---|---|---|
| HHS \| NIH \| National Institute of Allergy and Infectious Diseases (NIAID) | R01AI173686 | Catherine A. Wakeman |
| HHS \| NIH \| National Institute of General Medical Sciences (NIGMS) | R15GM128072 | Catherine A. Wakeman |

## AUTHOR CONTRIBUTIONS

Hafij Al Mahmud, Conceptualization, Data curation, Formal analysis, Methodology, Writing – original draft, Writing – review and editing | Randy Garcia, Data curation, Writing – review and editing | Alexsis Garcia, Data curation, Writing – review and editing | Jiwasmika Baishya, Data curation, Writing – review and editing | Catherine A. Wakeman, Conceptualization, Funding acquisition, Investigation, Methodology, Project administration, Resources, Supervision, Writing – review and editing

## ADDITIONAL FILES

The following material is available online.

### Supplemental Material

**Fig. S1 (Spectrum04226-23-s0001.docx).** Rescue of pyrimidine-deficient mutants of *P. aeruginosa* through Uridine-5′-monophosphate.

### Open Peer Review

**PEER REVIEW HISTORY (review-history.pdf).** An accounting of the reviewer comments and feedback.

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
