## [Reviewer comments · Microbiology Spectrum]

Microbiology Spectrum

Rescue of pyrimidine-defective *Pseudomonas aeruginosa* through metabolic complementation

Hafij Al Mahmud, Randy Garcia, Alexis Garcia, Jiwasnika Baishya, and Catherine Wakeman

Corresponding Author(s): Catherine Wakeman, Texas Tech University

Review Timeline:

Submission Date:	December 18, 2023
Editorial Decision:	February 7, 2024
Revision Received:	April 24, 2024
Editorial Decision:	May 22, 2024
Revision Received:	June 23, 2024
Accepted:	June 28, 2024

Editor: Justin Kaspar

Reviewer(s): Disclosure of reviewer identity is with reference to reviewer comments included in decision letter(s). The following individuals involved in review of your submission have agreed to reveal their identity: Robert P Smith (Reviewer #2)

Transaction Report:

DOI: <https://doi.org/10.1128/spectrum.04226-23>

Re: Spectrum04226-23 (Pyrimidine exchange between *Pseudomonas aeruginosa* and *Staphylococcus aureus*)

Dear Dr. Catherine Ann Wakeman:

Thank you for the privilege of reviewing your work. Below you will find my comments, instructions from the Spectrum editorial office, and the reviewer comments.

The manuscript has been evaluated by two experts in the field, and both have reached similar conclusions. Both reviewers, as well as my evaluation of the manuscript, agree that the overall conclusions presented do not align with the supplied data regarding the role of eDNA in complementation of pyrimidine-deficient strains of *Pseudomonas*. As you will read in their comments below, both reviewers felt other alternatives, such as intermediates of pyrimidine synthesis produced by *Staphylococcus*, were not rigorously tested and the data regarding complementation provided by eDNA on its own was relatively weak. Further experiments to support the main conclusion of the manuscript will be required.

Please note that while I have chosen not to reject the manuscript at this time, a substantial overhaul of the manuscript that includes additional evidence as the reviewers describe will be required for a revision to be considered at ASM Spectrum. I will be happy to extend the 60 day deadline for a resubmission so that these changes can be incorporated.

Revision Guidelines

Sincerely,
Justin Kaspar
Editor
Microbiology Spectrum

Reviewer #1 (Comments for the Author):

This manuscript explores an initial observation that the growth of *Pseudomonas aeruginosa* pyrimidine auxotrophic mutants could be rescued by co-culture with *Staphylococcus aureus*. The authors measured eDNA in these co-cultures, attempted to rescue growth by supplementation with exogenous DNA, and measured bacterial growth when direct contact was prevented. Although the initial observation is interesting, the authors' conclusions that secreted eDNA mediates the growth rescue are not supported by their data. While eDNA was detected in the co-cultures, supplementation with exogenous DNA only caused a slight increase in growth of the pyrimidine mutants. There is also an overall lack of rigor in data interpretation and strain validation.

Specific comments:

Were the pyrimidine auxotrophs validated by growth in minimal medium with and without pyrimidine supplementation? Can the mutants be complemented?

Does co-culture of pyrimidine mutants with wild-type PA14 restore growth? Is this response specific to *S. aureus* co-culture?

Ln 182 - 'less virulent' does not seem like an appropriate term here, since these experiments were all performed in the absence of host cells

Ln 183-184 - eDNA was not examined in Fig 1, so conclusions about its role in rescue of the PA14 pyrimidine auxotrophs are not justified

Fig 2 - Was eDNA quantification normalized to CFU/ml for each culture? Fig 1 shows that the pyrimidine mutants have a growth defect. Less dense cultures would be expected to contain less eDNA.

Ln 192-194 - Statistics for this comparison (JE2 and pyrimidine mutants) are not shown in the figure.

Ln 194-196 - It's not clear how JE2 having more eDNA than the PA mutants validates the PA mutants' auxotrophy. The conclusion about DNA secretion vs cell lysis is also not clear.

Ln 210-211 - These two figures show that co-culture with JE2 can rescue growth of PA14 pyrimidine auxotrophs and that eDNA can be detected to varying extents in these cultures. eDNA as the cause of the growth rescue has not been established.

Fig 3 and 4 - These data seem to argue against the conclusion that eDNA mediates rescue of the pyrimidine auxotrophs. Very small growth differences are observed with eDNA supplementation and growth is not restored to the level of WT.

Fig 5 legend (Ln 324-331) does not indicate the difference between panels A and B. Without this information, it is impossible to interpret the results. Both panels show co-cultures between pyrB mutant and JE2, but they show different results.

Simplifying the abbreviations for *S. aureus* and *P. aeruginosa* by using just one abbreviation each (either SA/PA or the strain names JE2/PA14) would help the reader follow the manuscript.

Reviewer #2 (Comments for the Author):

In this manuscript, Al Mahmud et al., present evidence that pyrimidines produced by *S. aureus* (SA) can rescue *P. aeruginosa* (PA) that lacks the ability to perform pyrimidine synthesis. The authors use a series of PA mutants created using a transposon each lacking a unique gene in pyrimidine synthesis. They then show that, alone, these mutant strains survive, but do not grow, in minimal medium. However, complementation with SA or herring DNA returns growth to the mutants. The authors then show that there is a likely a spatial component to the exchange. Taken together, the manuscript adds to our understanding of how PA and SA interact.

Overall, the manuscript is well written. The figures and figure legends are very clear, which I really appreciate. However, I am concerned that that it more than just pyrimidines that are involved in the rescue. It could be an entirely different substrate that feeds into pyrimidine synthesis that is overproduced by SA. I think that to truly show that it is pyrimidines that are rescuing PA when co-culture with SA, the authors need to perform some important control experiments.

Specific points:

1) The introduction is really short and misses a lot of interactions that have been previously shown between these two pathogens, both antagonistic and mutualistic. These should be summarized. Furthermore, because this manuscript involves

several genes and (presumably) nitrogenous base scavenging, a brief paragraph on purine/pyrimidine synthesis and salvage is warranted to put the mutants (pyrF, etc.) in the greater context of the pathways in which they are found.

2) Throughout the first part of the results and discuss, the reader is made to think that eDNA is being secreted by and that is rescuing PA. But SA could also produce metabolites that feed into pyrimidine synthesis, which may be rescuing PA. If it was simply a matter of providing pyrimidines, then shouldn't all of the mutants be rescued more or less in the exact same way. That is, in figure 3, shouldn't all of the mutants behave, for example, like pyrF? I understand that this data is not in the presence of SA but because cytosine enters pyrimidine synthesis after purF (and thus all of the other enzymes that they have knocked out) and thymine enters via a route that doesn't involve the pyr enzymes, shouldn't they all behave the same? This to me is very puzzling... If it was simply just pyrimidines, could the PA mutants be rescued with supplementation with pyrimidines? Have the authors tried these experiments (as opposed to herring DNA)? To me, that would provide more evidence that only pyrimidines are responsible for rescue.

3) Another way to provide support for pyrimidines in rescue in acquiring SA that lacks a gene in pyrimidine synthesis early on in the pathway (right after the prpp autoregulation edge in the pathway). Co-culture with this strain should not lead to growth of either strain. That would provide more conclusive evidence that it is pyrimidines (nitrogenous bases or the deoxy form) that is rescuing PA.

4) Can PA import/salvage deoxy nucleotides? Or are only nitrogenous bases imported?

5) The authors use the dye picogreen to bind to eDNA. Based on their methods, I think that the picogreen assay was performed in the presence of intact (and maybe live) cells. How can the authors rule out that picogreen is not binding to DNA internal the cells. If I were to approach his problem, I would filter the medium to remove cells (with e.g., 0.45 um filter, which should remove cells but leave eDNA) and then stain the cell free medium. To me, this would ensure that this DNA is outside of the cells.

6) Will cell free but SA conditioned medium rescue PA? If it does, then I think that the authors can certainly strengthen their claim that something secreted is rescuing PA.

Overall, I think that that authors have a really good idea, and if it is supported by additional, critical control experiments above, would certainly have an impact on the field. But without the experiments above, while it could be pyrimidines that are exchanged, it might also be a number of other things that are rescuing the PA mutant strains.

We really appreciate the reviewer's for their valuable feedback, comments and suggestions. We tried to address all the concerns raised by the reviewer's. We believe their feedback increased the quality of this manuscript.

In the response section, the **text highlighted in green** is presented here as a reference to the direct intext revision in the **clean** manuscript.

Thanks so much!

Sincerely,

Hafij Al Mahmud

Reviewer #1 (Comments for the Author):

This manuscript explores an initial observation that the growth of *Pseudomonas aeruginosa* pyrimidine auxotrophic mutants could be rescued by co-culture with *Staphylococcus aureus*. The authors measured eDNA in these co-cultures, attempted to rescue growth by supplementation with exogenous DNA, and measured bacterial growth when direct contact was prevented. Although the initial observation is interesting, the authors' conclusions that secreted eDNA mediates the growth rescue are not supported by their data. While eDNA was detected in the co-cultures, supplementation with exogenous DNA only caused a slight increase in growth of the pyrimidine mutants. There is also an overall lack of rigor in data interpretation and strain validation.

Response: Thanks for highlighting this issue. Based on the reviewer's comment, we revisited our data, and also, based on the new additional data, we agree with the reviewer's concern that eDNA would be a minor contributor to pyrimidine complementation. We have revised the whole manuscript accordingly, including our concluding remarks in lines 380-387

"Both metabolically active and lysed cells may facilitate the pyrimidine complementation in neighboring pathogens. Importantly, secreted eDNA is a minor contributor in pyrimidine complementation towards rescuing the pyrimidine-defective PA, but neighboring cells (SA) are more proficient for pyrimidine exchange via intricate cell-to-cell contact mechanism. Apart from eDNA, secreted nucleotides, such as UMP, a precursor for pyrimidine nucleotide, may also rescue pyrimidine defective mutants. The role of secreted nucleosides and nitrogenous bases in this complementation requires further investigation."

Also, we have validated one of the mutant strains, *pyrB::tn*, through whole genome sequencing (Please find the edits in line 115 and 263).

Specific comments:

Were the pyrimidine auxotrophs validated by growth in minimal medium with and without pyrimidine supplementation? Can the mutants be complemented?

Response: We really appreciate you pointing out this critical issue. Yes, the pyrimidine-defective mutants are defective in growth in RPMI (+1% casamino acid) minimal medium compared to wild-type PA14 (*pyr::tn* monoculture growth in Figures 1, 2, 4).

According to your comment and suggestion, in addition to eDNA, we conducted a chemical complementation experiment using UMP, a precursor for pyrimidine nucleotide synthesis and a product of de novo pyrimidine biosynthesis. We found that UMP can also significantly rescue the growth of a pyrimidine-defective mutant of PA.

Please find the added data in lines 261-263 and Supplemental Figure 1

“Furthermore, we confirmed the *pyrB::tn* mutant to be truly deficient in pyrimidine biosynthesis using chemical complementation with uridine-5'-monophosphate (Supplemental Figure 1) as well as whole genome sequencing (data not shown)”

Because of time limitations, we could not include all the pyrimidine-defective mutants for the extended experiments.

Does co-culture of pyrimidine mutants with wild-type PA14 restore growth? Is this response specific to *S. aureus* co-culture?

Response: To address your feedback, we have conducted a co-culture experiment using *pyrB::tn* as a representative pyrimidine-defective mutant with wild-type PA14, *Enterococcus faecalis*, and *Acinetobacter baumannii*. Our data showed that the growth of *pyrB::tn* can be rescued with any of these organisms, which means the rescue is not specific to SA.

Please see the results in lines 256-273 and Figures 2A and B

“Further, we wanted to explore whether the observed metabolic complementation of pyrimidine-defective PA is restricted to SA only or not. To this extent, we conducted similar experiments using AB, a representative gram-negative pathogen, and EF, a representative gram-positive pathogen. Both AB and EF are associated with different life-threatening infections.^{28,29} Pyrimidine-defective mutant *pyrB::tn* was selected as a representative auxotrophic mutant for this experiment. Furthermore, we confirmed the *pyrB::tn* mutant to be truly deficient in pyrimidine biosynthesis using chemical complementation with uridine-5'-monophosphate (Supplemental Figure 1) as well as whole genome sequencing (data not shown). Like earlier mixed culture experiments with JE2, the fitness of PA14 was found to be unaffected in mixed culture with AB and EF compared to monoculture. Whereas, compared to monoculture, a significant ($p < 0.0001$) rescue in *pyrB::tn* cells has been found in the presence of both AB and FE (Figure 2A). That tells us that the growth rescue of the pyrimidine-defective mutants is not restricted to SA or gram-negative or gram-positive pathogens. Neighboring pathogens with functional pyrimidine biosynthesis machinery may rescue pyrimidine-defective mutants of PA in infection sites. Furthermore, both the wildtype PA14 and *pyrB::tn* mutant were found to be slightly competitive ($p < 0.05$) against AB, whereas EF was found to be unresponsive to PA14 and *pyrB::tn* mediated killing in mixed culture (Figure 2A). That tells that the molecules responsible for the pyrimidine complementation may come from metabolically active or lysed cells.”

Ln 182 - 'less virulent' does not seem like an appropriate term here, since these experiments were all performed in the absence of host cells

Response: Thanks so much for your feedback. We deleted 'virulent' from the statement, and now the statement would be “mutants may be attributed to the fact that the auxotrophic strains are less competitive against JE2 compared to PA14” in lines 251-252

Ln 183-184 - eDNA was not examined in Fig 1, so conclusions about its role in rescue of the PA14 pyrimidine auxotrophs are not justified

Response: We appreciate your feedback. We agree with the point that it would be too early to claim eDNA as the rescuer. Based on reviewers comment, we revised the statements as follows in lines 240-245.

“These data demonstrate that JE2 can rescue the pyrimidine-deficient PA in mixed cultures. This rescue might be achieved through release of eDNA, nucleosides, or nitrogenous bases. The release of eDNA or other secretory molecules in the liquid cultures may be a result of cellular degradation following lysis and/or secretion by metabolically active bacterial cells. To identify the possible source of these molecules, either from lysis or secretion of metabolically active bacteria, we estimated the fitness of JE2 in mono and mixed cultures (Figure 1C).”

Fig 2 - Was eDNA quantification normalized to CFU/ml for each culture? Fig 1 shows that the pyrimidine mutants have a growth defect. Less dense cultures would be expected to contain less eDNA.

Response: We started this experiment with similar cell density, but we did not normalize eDNA to CFU/ml for each culture. Our main goal of this experiment was to measure the secreted eDNA in the culture over time. We agree that the defective growth of the PA mutant would produce less eDNA. Actually, we wanted to explore this particular phenomenon with this experiment to show that these mutants are incapable of producing enough pyrimidine in monoculture, therefore creating less eDNA, which in turn makes them growth defective as pyrimidines are essential for their growth and development.

Ln 192-194 - Statistics for this comparison (JE2 and pyrimidine mutants) are not shown in the figure.

Response: We appreciate your feedback. We have revised the figure 3 accordingly.

Ln 194-196 - It's not clear how JE2 having more eDNA than the PA mutants validates the PA mutants' auxotrophy. The conclusion about DNA secretion vs cell lysis is also not clear.

Response: Thanks so much for your comment. We agree that, without further studies, it would not be justifiable to claim the PA mutant's auxotrophy just by comparing the eDNA in JE2 monoculture. Therefore, we revised the manuscript to address this concern and tried to make it clearer, as follows: lines 287-293.

“Our results depict that the presence of eDNA in JE2 monoculture is higher than in PA's pyrimidine-defective mutants (pyrB::tn; $p > 0.05$, pyrD::tn; $p < 0.05$, pyrE::tn; $p < 0.05$, pyrF::tn; $p > 0.05$) which might be attributed to the low growth of the defective mutants in monoculture (Figure 3). Different microorganisms, including bacteria, may release eDNA in the environment through various mechanisms, such as active secretion by metabolically active cells, by membrane vesicles, or following cell lysis.”

Ln 210-211 - These two figures show that co-culture with JE2 can rescue growth of PA14 pyrimidine auxotrophs and that eDNA can be detected to varying extents in these cultures. eDNA as the cause of the growth rescue has not been established.

Response: We really appreciate you pointing out this critical issue. Based on the reviewer's comment we revisited our data and also based on the new additional data we agree with

the reviewer's concern that eDNA would be a minor contributor in pyrimidine complementation. We have revised the whole manuscript accordingly including our concluding remarks in lines 380-387.

"Both metabolically active and lysed cells may facilitate the pyrimidine complementation in neighboring pathogens. Importantly, secreted eDNA is a minor contributor in pyrimidine complementation towards rescuing the pyrimidine-defective PA, but neighboring cells (SA) are more proficient for pyrimidine exchange via intricate cell-to-cell contact mechanism. Apart from eDNA, secreted nucleotides, such as UMP, a precursor for pyrimidine nucleotide, may also rescue pyrimidine defective mutants. The role of secreted nucleosides and nitrogenous bases in this complementation requires further investigation."

In addition to eDNA, we conducted a new chemical complementation experiment using UMP, a precursor for the pyrimidine nucleotide and a product of de novo pyrimidine biosynthesis. We found that UMP can significantly rescue the growth of a pyrimidine-defective mutant of PA as well. That means, secretory nucleotides can also play a role in pyrimidine exchange.

Please find the added data in lines 261-263 and Supplemental Figure 1A

"Furthermore, we confirmed the pyrB::tn mutant to be truly deficient in pyrimidine biosynthesis using chemical complementation with uridine-5'-monophosphate (Supplemental Figure 1) as well as whole genome sequencing (data not shown)"

Fig 3 and 4 - These data seem to argue against the conclusion that eDNA mediates rescue of the pyrimidine auxotrophs. Very small growth differences are observed with eDNA supplementation and growth is not restored to the level of WT.

Response: We really appreciate you pointing out this critical issue. Based on the reviewer's comment we revisited our data and also based on the new additional data we agree with the reviewer's concern that eDNA would be a minor contributor in pyrimidine complementation. We have revised the whole manuscript accordingly including our concluding remarks in lines 380-387.

"Both metabolically active and lysed cells may facilitate the pyrimidine complementation in neighboring pathogens. Importantly, secreted eDNA is a minor contributor in pyrimidine complementation towards rescuing the pyrimidine-defective PA, but neighboring cells (SA) are more proficient for pyrimidine exchange via intricate cell-to-cell contact mechanism. Apart from eDNA, secreted nucleotides, such as UMP, a precursor for pyrimidine nucleotide, may also rescue pyrimidine defective mutants. The role of secreted nucleosides and nitrogenous bases in this complementation requires further investigation."

Fig 5 legend (ln 324-331) does not indicate the difference between panels A and B. Without this information, it is impossible to interpret the results. Both panels show co-cultures between pyrB mutant and JE2, but they show different results.

Respond: We appreciate your feedback. We have revised the figure legend accordingly. Please find the correction in revised figure 6.

Simplifying the abbreviations for *S. aureus* and *P. aeruginosa* by using just one abbreviation each (either SA/PA or the strain names JE2/PA14) would help the reader follow the manuscript.

Respond: We appreciate your feedback. We have revised the manuscript according to the reviewer's comments. Throughout the manuscript, PA and SA were used to refer to *P. aeruginosa* and *S. aureus*. PA14 denotes wild-type PA and JE2 denotes wild-type JE2

Reviewer #2 (Comments for the Author):

In this manuscript, Al Mahmud et al., present evidence that pyrimidines produced by *S. aureus* (SA) can rescue *P. aeruginosa* (PA) that lacks the ability to perform pyrimidine synthesis. The authors use a series of PA mutants created using a transposon each lacking a unique gene in pyrimidine synthesis. They then show that, alone, these mutant strains survive, but do not grow, in minimal medium. However, complementation with SA or herring DNA returns growth to the mutants. The authors then show that there is a likely a spatial component to the exchange. Taken together, the manuscript adds to our understanding of how PA and SA interact.

Overall, the manuscript is well written. The figures and figure legends are very clear, which I really appreciate. However, I am concerned that that it more than just pyrimidines that are involved in the rescue. It could be an entirely different substrate that feeds into pyrimidine synthesis that is overproduced by SA. I think that to truly show that it is pyrimidines that are rescuing PA when co-culture with SA, the authors need to perform some important control experiments.

Response: We appreciate your feedback. To address this concern, we conducted a new chemical complementation experiment using UMP, a precursor for the pyrimidine nucleotide and a product of de novo pyrimidine biosynthesis. We found that UMP can significantly rescue the growth of a pyrimidine-defective mutant of PA as well. That means, secretory nucleotides can also play a role in pyrimidine exchange.

Please find the added data in lines 261-263 and Supplemental Figure 1

“Furthermore, we confirmed the *pyrB::tn* mutant to be truly deficient in pyrimidine biosynthesis using chemical complementation with uridine-5'-monophosphate (Supplemental Figure 1) as well as whole genome sequencing (data not shown).”

Specific points:

1) The introduction is really short and misses a lot of interactions that have been previously shown between these two pathogens, both antagonistic and mutualistic. These should be summarized. Furthermore, because this manuscript involves several genes and (presumably) nitrogenous base scavenging, a brief paragraph on purine/pyrimidine synthesis and salvage is warranted to put the mutants (pyrF, etc.) in the greater context of the pathways in which they are found.

Response: We really appreciate your feedback, which we believe increased our lit review quality. We incorporated these important studies in lines 56-60, 62-70, and 81-91.

“For example, iron depletion in co-culture may increase the lysis of SA by PA through secreted 2-alkyl-4(1H)-quinolones.⁴ Bacterial co-culture on human bronchial epithelial cell monolayers showed PA drives SA metabolism from aerobic respiration to fermentation and eventually kills SA by secreting siderophores or 2-heptyl-4-hydroxyquinoline N-oxide (HQNO).⁵”

For example, PA isolated from coinfecting patients is found to be less competitive against SA; in fact, PA with mucoid phenotype would become severely inactive against SA and would reside within the infection site together. Alginate-producing mucoid strains of PA downregulate the synthesis of different virulence factors essential for the killing of SA.⁷ In the polymicrobial lung infection model, virulence factors secreted by SA are found to be helpful in the proliferation, spread, and pathogenicity of gram-negative pathogens like PA through compromising host immunity.⁸ In addition, host immune proteins like calprotectin may facilitate the co-colonization of these two classical competitors, PA and SA, in cystic fibrosis lung.⁹ “

“Pathogenic bacteria rely on de novo nucleotide biosynthesis to initiate infection, survive, and be virulent. Regulators of this process have been shown to be essential in regulating the production of virulence factors.^{14,15} Synthesis or acquisition of purines and pyrimidines is essential for cellular functions and the reproduction of microorganisms.¹⁴ For example, pyrimidine biosynthetic genes are essential for PA to grow well in the CF-infected lung environment.¹⁶ In de novo purine biosynthesis, precursor molecule 5-phosphoribosyl- α -1-pyrophosphate (PRPP) is converted into the final product inosine-5'-monophosphate (IMP) by the action of different enzymes, encoded by genes such as purF, purD, purN, purT, purS, purQ, purL, purM, purK, purE, purC, purB, and purH, etc.¹⁴ Similarly, two methods are employed by organisms for obtaining pyrimidines: de novo synthesis, which is a universal process consisting of six consecutive enzyme reactions, or a salvage pathway.¹⁷ In contrast, in de novo pyrimidine biosynthesis, the precursor molecule carbamoyl phosphate (CP) is converted into the final product uridine-5'-monophosphate (UMP) by the action of different enzymes, namely carbamoyl phosphate synthetase, aspartate transcarbamylase, dihydroorotase, dihydroorotate dehydrogenase, orotate phosphoribosyltransferase, and orotidylate decarboxylase.^{18,19} These enzymes are encoded by six unlinked genes, namely pyrB, pyrC, pyrK, pyrD, pyrE, and pyrF, etc.¹⁴ Finally, the precursor molecule UMP can be converted to different pyrimidines, such as UDP, UTP, CTP, etc., by the action of other enzymes. Apart from these, UMP can be produced from pyrimidine bases and nucleosides by enzymatic action through salvage pathway.¹⁷ ”

2) Throughout the first part of the results and discuss, the reader is made to think that eDNA is being secreted by and that is rescuing PA. But SA could also produce metabolites that feed into pyrimidine synthesis, which may be rescuing PA. If it was simply a matter of providing pyrimidines, then shouldn't all of the mutants be rescued more or less in the exact same way.

That is, in figure 3, shouldn't all of the mutants behave, for example, like pyrF? I understand that this data is not in the presence of SA but because cytosine enters pyrimidine synthesis after purF (and thus all of the other enzymes that they have knocked out) and thymine enters via a route that doesn't involve the pyr enzymes, shouldn't they all behave the same? This to me is very puzzling... If it was simply just pyrimidines, could the PA mutants be rescued with supplementation with pyrimidines? Have the authors tried these experiments (as opposed to herring DNA)? To me, that would provide more evidence that only pyrimidines are responsible for rescue.

Response: We really appreciate you pointing out this critical issue. Based on the reviewer's comment we revisited our data and also based on the new additional data we agree with the reviewer's concern that eDNA would be a minor contributor in pyrimidine complementation. We have revised the whole manuscript accordingly including our concluding remarks in lines 380-387.

"Both metabolically active and lysed cells may facilitate the pyrimidine complementation in neighboring pathogens. Importantly, secreted eDNA is a minor contributor in pyrimidine complementation towards rescuing the pyrimidine-defective PA, but neighboring cells (SA) are more proficient for pyrimidine exchange via intricate cell-to-cell contact mechanism. Apart from eDNA, secreted nucleotides, such as UMP, a precursor for pyrimidine nucleotide, may also rescue pyrimidine defective mutants. The role of secreted nucleosides and nitrogenous bases in this complementation requires further investigation."

In addition to eDNA, we conducted a new chemical complementation experiment using UMP, a precursor for the pyrimidine nucleotide and a product of de novo pyrimidine biosynthesis. We found that UMP can significantly rescue the growth of a pyrimidine-defective mutant of PA as well.

Please find the added data in lines 261-263 and Supplemental Figure 1

"Furthermore, we confirmed the pyrB::tn mutant to be truly deficient in pyrimidine biosynthesis using chemical complementation with uridine-5'-monophosphate (Supplemental Figure 1) as well as whole genome sequencing (data not shown)"

We agree that it is interesting that we are seeing varying rescues between different mutants. Because of time limitations, we could not include all the pyrimidine-defective mutants for the extended experiments. However, we think it would be exciting to explore this phenomenon further in future studies.

3) Another way to provide support for pyrimidines in rescue in acquiring SA that lacks a gene in pyrimidine synthesis early on in the pathway (right after the prpp autoregulation edge in the pathway). Co-culture with this strain should not lead to growth of either strain. That would provide more conclusive evidence that it is pyrimidines (nitrogenous bases or the deoxy form) that is rescuing PA.

Response: We agree with this comment. We also wanted to add this important control in this study, but unfortunately, we do not have such a pyrimidine-defective mutant in our lab now. However, it would be interesting to conduct a controlled study in the future.

4) Can PA import/salvage deoxy nucleotides? Or are only nitrogenous bases imported?

Response: Thanks so much for your comment. Due to time constraints, we could not test nitrogenous bases. However, we conducted experiments using nucleotide bases like UMP, which tells us PA may import/salvage nucleotides. Also, as the PA strain was defective for only one gene in the *de novo* pathway, we assume they might also be able to salvage nitrogenous bases. But we believe this could be an interesting study in the future.

5) The authors use the dye picogreen to bind to eDNA. Based on their methods, I think that the picogreen assay was performed in the presence of intact (and maybe live) cells. How can the authors rule out that picogreen is not binding to DNA internal the cells. If I were to approach his problem, I would filter the medium to remove cells (with e.g., 0.45 um filter, which should remove cells but leave eDNA) and then stain the cell free medium. To me, this would ensure that this DNA is outside of the cells.

Response: Thanks so much for pointing out this critical factor. The reason we did use this dye in a bacterial culture is that, usually, this cannot penetrate the bacterial membrane, and therefore, it would not bind with cellular DNA. Sorry for not clarified this in the manuscript earlier. Thus, we added the following statement in the manuscript in lines 141-142.

"Generally, picogreen is incapable of penetrating bacterial cell membranes; therefore, it would only bind with the eDNA in the bacterial culture." ([10.4236/jst.2016.63003](https://doi.org/10.4236/jst.2016.63003))

Also, "The standard curve was generated by measuring the fluorescent intensities of known concentrations (0 to 900 µg/ml) of eDNA in respective bacterial monocultures in RPMI media." (lines 147-149)

Without eDNA in bacterial culture, we did not see any fluorescence intensity in the sample, which also demonstrate that picogreen does not penetrate bacterial cell membranes.

6) Will cell free but SA conditioned medium rescue PA? If it does, then I think that the authors can certainly strengthen their claim that something secreted is rescuing PA.

Response: We appreciate your feedback. According to your suggestion, we conducted an experiment to determine whether or not SA cell-free supernatant can rescue pyrimidine defective PA mutants. We found that similar to eDNA, SA cell-free supernatant can also rescue pyrimidine-defective PA, but only partially, compared to the complete rescue found in mixed culture with JE2. Please find the results in Ln 364-372 and Figure 6C.

"In this stage, we also wanted to see if the JE2 secretion is capable of rescuing the pyrimidine-defective mutants in the absence of metabolically active cells. To achieve this, we filter sterilized JE2 supernatants and then culture PA14 and pyrB:tn cells with or without the presence of JE2 supernatants. Our data

showed that JE2 supernatant does not affect the fitness of PA14, but it significantly ($p < 0.05$) increases the fitness of *pyrB::tn* cells (Figure 6C). Interestingly, similar to eDNA, the rescue of pyrimidine-defective mutants mediated by JE2 supernatant is only partial compared to the complete rescue observed in mixed culture with JE2. JE2 supernatant may contain eDNA, pyrimidine nucleotides, pyrimidine nucleosides, or nitrogenous bases that can complement the pyrimidine deficiencies in auxotrophic bacteria.”

Overall, I think that that authors have a really good idea, and if it is supported by additional, critical control experiments above, would certainly have an impact on the field. But without the experiments above, while it could be pyrimidines that are exchanged, it might also be a number of other things that are rescuing the PA mutant strains.

Re: Spectrum04226-23R1 (Pyrimidine exchange between *Pseudomonas aeruginosa* and *Staphylococcus aureus*)

Dear Dr. Catherine Ann Wakeman:

Thank you for the privilege of reviewing your work. Below you will find my comments, instructions from the Spectrum editorial office, and the reviewer comments.

The referenced manuscript was resent to the two original reviewers upon receipt, and as you will see, both reviewers commended the revision as much improved. Still, there are a few points as both reviewers note that need modification prior to acceptance. The first is the title of the manuscript - I do agree with reviewer 1 that based on the modifications, the title and other areas could be edited, as noted to reflect the restated conclusions. In addition, both reviewers pointed out line 298 as an area regarding metabolic activity of cells during eDNA secretion as an idea that should be amended. Both reviewers also supply other comments that can be addressed (see bottom of this email).

Revision Guidelines

Sincerely,
Justin Kaspar
Editor
Microbiology Spectrum

Reviewer #1 (Comments for the Author):

Overall, the authors have addressed the reviewer comments well and the manuscript's conclusions are appropriate. A few minor concerns should still be addressed, however.

Specific comments:

The conclusions of the manuscript have been modified to include the possibility that precursor molecules or other metabolites (besides pyrimidines) rescue the auxotrophs. In light of these changes, references to 'pyrimidine exchange' or 'pyrimidine complementation' in the Title, Abstract (Ln 34 and 43) and Conclusions (Ln 384-389) should also be modified.

Ln 244-256. It's not clear how the fitness of JE2 in mono-culture vs co-culture informs about the degree of cell lysis. Both mono and mixed cultures likely have lysed cells. Conclusions should be modified (Ln 254-256).

Ln 260. Write out abbreviations here for AB and EF.

Ln 280-282. Conclusions should be modified here. It's not clear how these transwell experiments show that the complementation is due to metabolically active cells and not cell lysis. These experiments show that a PA14 diffusible factor is responsible for the complementation and does not require cell-cell contact.

Ln 298-300. Conclusions should be modified here. The eDNA quantification simply shows that eDNA can be detected in culture supernatants.

Reviewer #2 (Comments for the Author):

I would like to thank the authors for addressing all of my concerns from the previous review. To me, their experiments and changes have improved the manuscript.

I have a few minor things that I feel should be addressed.

Supplementary Flg 1 - is this mg/mL of UMP? or ug/mL of UMP?

line 265 - sequencing comment should be moved to the methods because it is really not a result.

line 298 - I dont understand how the authors claim that the cells have to be metabolically active to secrete eDNA. I dont think that this has been shown... instead what has been shown in the a) eDNA exists, and b) it is more the wildtype strains. Any claim on metabolically active/inactive would have to be verified by measuring the metabolism of the cells. as this would open up another can of worms so to speak, i would recommend that the authors state what their results can support as noted above.

We really appreciate the reviewers' valuable feedback. We tried to address all the concerns raised by the reviewers. We believe their feedback has enhanced the quality of this manuscript.

In the response section, the **text highlighted in green** refers to the direct intext revision, and the **text highlighted in red** represents deleted statements in the **clean** manuscript.

Thanks so much!

Sincerely,

Hafij Al Mahmud

Reviewer #1 (Comments for the Author):

Overall, the authors have addressed the reviewer comments well and the manuscript's conclusions are appropriate. A few minor concerns should still be addressed, however.

Specific comments:

The conclusions of the manuscript have been modified to include the possibility that precursor molecules or other metabolites (besides pyrimidines) rescue the auxotrophs. In light of these changes, references to 'pyrimidine exchange' or 'pyrimidine complementation' in the Title, Abstract (In 34 and 43) and Conclusions (In 384-389) should also be modified.

Response: We appreciate your feedback. We revised the title to: "Rescue of pyrimidine-defective *Pseudomonas aeruginosa* through metabolic complementation."

In addition, we revised the mentioned statements as follows in lines 34, 42, 381-385 respectively.

Our data further highlights the importance of cell-to-cell contact for effective and increased **metabolic** complementation.

Furthermore, our findings highlight the mechanisms involved in **metabolic** exchange, emphasizing the importance of cell-to-cell contact.

Both metabolically active and lysed cells may facilitate the **metabolic** complementation in neighboring pathogens. Importantly, secreted eDNA is a minor contributor in **metabolic** complementation towards rescuing the pyrimidine-defective PA, but neighboring cells (SA) are more proficient for **metabolic** exchange via cell-to-cell contact mechanism.

Ln 244-256. It's not clear how the fitness of JE2 in mono-culture vs co-culture informs about the degree of cell lysis. Both mono and mixed cultures likely have lysed cells. Conclusions should be modified (Ln 254-256).

Response: Thanks so much for pointing out this important issue. We have revised the statement in Ln 244-256, and the conclusion is as follows.

We replaced the statement "To identify the possible source of these molecules, either from lysis or secretion of metabolically active bacteria, we estimated the fitness of JE2 in mono and mixed cultures (Figure 1C)" with "Furthermore, we estimated the fitness of JE2 in mono and mixed cultures (Figure 1C)" in lines 245-246

We replaced the following statement "Furthermore, these data demonstrate that the secretory rescuer molecules in mixed cultures may be significantly contributed by metabolically active JE2 apart from degraded JE2 cells." with "Overall, the metabolites mediating interspecies complementation may, at least in part, be released from lysed cell populations." In lines 253-255

Ln 260. Write out abbreviations here for AB and EF.

Response: We revised the sentence accordingly in line 258-260.

To this extent, we conducted similar experiments using *Acinetobacter baumannii* (AB), a representative gram-negative pathogen, and *Enterococcus faecalis* (EF), a representative gram-positive pathogen.

Ln 280-282. Conclusions should be modified here. It's not clear how these transwell experiments show that the complementation is due to metabolically active cells and not cell lysis. These experiments show that a PA14 diffusible factor is responsible for the complementation and does not require cell-cell contact.

Response: Thanks so much for your feedback. We revised the conclusion accordingly in Lines 278-280

In summary, these data emphasize that metabolites secreted by wild-type PA14 can complement pyrimidine deficiency in defective mutants. Additionally, intraspecies complementation can be achieved without cell-to-cell contact.

Ln 298-300. Conclusions should be modified here. The eDNA quantification simply shows that eDNA can be detected in culture supernatants.

Response: We appreciate your feedback. Please find the revised conclusion in lines 295-298.

Overall, the data from **Figure 3** suggests that eDNA is present in the culture supernatant and implies that microbe-derived eDNA also contributes to the overall eDNA present in the infection niches. In addition, the concentration of eDNA in wild-type culture supernatant is higher in comparison to defective mutants.

Reviewer #2 (Comments for the Author):

I would like to thank the authors for addressing all of my concerns from the previous review. To me, their experiments and changes have improved the manuscript. I have a few minor things that I feel should be addressed.

Supplementary Flg 1 - is this mg/mL of UMP? or ug/mL of UMP?

Response: Thank you. In supplementary fig1, the concentration of UMP was in mg/ml.

line 265 - sequencing comment should be moved to the methods because it is really not a result.

Response: We appreciate your feedback. We deleted the sequencing comment from the results and moved that to the method section in lines 114-117

One of the mutants, *pyrB::tn*, was confirmed via whole genome sequencing performed by Plasmidsaurus. We acquired the FASTA sequence data of the *pyrB::tn* strain from Plasmidsaurus. This sequence was then aligned with the genome of the wild-type strain PA14 using Mauve, confirming the presence of a transposon insert in the *pyrB* gene of the mutant strain.

line 298 - I dont understand how the authors claim that the cells have to be metabolically active to secrete eDNA. I dont think that this has been shown... instead what has been shown in the a) eDNA exists, and b) it is more the wildtype strains. Any claim on metabolically active/inactive would have to be verified by measuring the metabolism of the cells. as this would open up another can of worms so to speak, i would recommend that the authors state what their results can support as noted above.

Response: We appreciate your response. Please find the revised conclusion in Ln 295-298

Overall, the data from **Figure 3** suggests that eDNA is present in the culture supernatant and implies that microbe-derived eDNA also contributes to the overall eDNA present in the infection niches. In addition, the concentration of eDNA in wild-type culture supernatant is higher in comparison to defective mutants.

Re: Spectrum04226-23R2 (Rescue of pyrimidine-defective *Pseudomonas aeruginosa* through metabolic complementation)

Dear Dr. Catherine Ann Wakeman:

Your manuscript has been accepted, and I am forwarding it to the ASM production staff for publication. Your paper will first be checked to make sure all elements meet the technical requirements. ASM staff will contact you if anything needs to be revised before copyediting and production can begin. Otherwise, you will be notified when your proofs are ready to be viewed.

Sincerely,
Justin Kaspar
Editor
Microbiology Spectrum